# Antioxidative Effect of *Chlorella Pyrenoidosa* Protein Hydrolysates and Their Application in Krill Oil-in-Water Emulsions

**DOI:** 10.3390/md20060345

**Published:** 2022-05-25

**Authors:** Yujia Liu, Yuli Qi, Qi Wang, Fawen Yin, Honglei Zhan, Han Wang, Bingnan Liu, Yoshimasa Nakamura, Jihui Wang

**Affiliations:** 1School of Biological Engineering, Dalian Polytechnic University, Dalian 116034, China; liuyuj@dlpu.edu.cn (Y.L.); qyl1123004@163.com (Y.Q.); wq2020ydsan@163.com (Q.W.); zhanhonglei121@163.com (H.Z.); hwang@dlpu.edu.cn (H.W.); lbnno158@foxmail.com (B.L.); 2School of Food Science and Technology, Dalian Polytechnic University, Dalian 116034, China; yinfawen198@126.com; 3National Engineering Research Center of Seafood, Dalian 116034, China; 4Graduate School of Environmental and Life Science, Okayama University, Okayama 700-8530, Japan; yossan@cc.okayama-u.ac.jp; 5School of Chemical Engineering and Energy Technology, Dongguan University of Technology, Dongguan 523808, China

**Keywords:** *Chlorella pyrenoidosa* protein hydrolysate, antioxidant activity, krill oil-in-water emulsion, lipid oxidation

## Abstract

*Chlorella pyrenoidosa* is an excellent source of protein, and in this research, we assessed the antioxidant and emulsifying effects of Chlorella protein hydrolysate (CPH) using neutral proteases and alkaline proteases, as well as the properties of CPH-derived krill oil-in-water (O/W) emulsions. The CPHs exhibited the ability to scavenge several kinds of free radicals, including 1,1-diphenyl-2-picrylhydrazyl (DPPH), O_2_^−^, hydroxyl, and ABTS. Additionally, the CPHs (5 mg/mL) scavenged approximately 100% of the DPPH and ABTS. The CPHs showed similar emulsifying activities to Tween 20 and excellent foaming activities (max FS 74%), which helped to stabilize the krill oil-in-water emulsion. Less than 10 mg/mL CPHs was able to form fresh krill oil-in-water emulsions; moreover, the CPHs (5 mg/mL) in a krill O/W emulsion were homogenous, opaque, and stable for at least 30 days. Based on their inhibitory effects on the peroxide value (POV) and thiobarbituric acid reactive substances (TRABS), the CPHs were found to be able to inhibit lipid oxidation in both emulsifying systems and krill O/W emulsions. Thus, the CPHs could improve superoxide dismutase (SOD) activities by 5- or 10-fold and decrease the high reactive oxygen species (ROS) level caused by the addition of H_2_O_2_ in vitro. In conclusion, health-promoting CPHs could be applied in krill oil-in-water emulsions as both emulsifiers and antioxidants, which could help to improve the oxidative and physical stability of emulsions.

## 1. Introduction

Antarctic krill (*Euphausia superba*), a well-known species, famous for its nutritional value, is considered one of the richest biomasses in the world, and the functional lipid content in krill has been proven to be approximately 12–50%. Therefore, Antarctic krill oil has an abundance of phospholipids, eicosapentaenoic acid (EPA), docosahexaenoic acid (DHA), and astaxanthin [1]. To date, Antarctic krill oil has been confirmed to have anti-inflammatory, anticancer, brain function-promoting, and cardiovascular disease-preventing functions [2]. DHA and EPA in krill oil are incorporated into phospholipids, and the bioavailability of krill oil improves as a result [3]. Developing different types of krill oil can provide more opportunities for Antarctic krill oil consumption.

An O/W emulsion is a classic oil form that is commonly used in food products. An O/W emulsion from oil can help change the water solubility and enrich the functional components of emulsions. In the food industry, O/W emulsions are usually stabilized by commercial emulsifiers such as Tween 20, Tween 80, and Span. However, O/W emulsions are easily oxidized because there are many components promoting lipid oxidation in the aqueous phase that require interfacial antioxidants for stable oxidation of emulsions. Additionally, some antioxidant proteins and protein hydrolysates can also be used as both emulsifiers and antioxidants for different kinds of oil and add to the nutritional value of emulsions. Shen et al. indicated that fish gelatin and milk protein Maillard reaction products can both stabilize and raise krill O/W emulsions’ stability [4]. Additionally, whey protein and its hydrolysates were shown to be able to stabilize medium-chain triglyceride oil emulsions [5]. Various other protein hydrolysates, such as porcine bone protein hydrolysates [6], soy protein hydrolysate [7], and rice dreg protein hydrolysate [8], have demonstrated their inhibition effects on lipid oxidation and different emulsifying properties. Finding additional stabilized emulsifiers and antioxidants from protein hydrolysates will help in the process of developing more stable and functional O/W emulsions; stable krill O/W emulsions, in particular, require further investigation.

*Chlorella pyrenoidosa* (*C. pyrenoidosa*) is a green alga with a long history that is widely used as a food supplement worldwide. Most *C. pyrenoidosa* is found growing in fresh water environments; however, some marine types of *C. pyrenoidosa* can also be cultured in seawater [9,10,11]. The marine species of *C. pyrenoidosa* could possibly decrease the large inputs needed to a greater extent than freshwater species, because of the requirement of fresh water. More than 50% of the *C. pyrenoidosa* cell content is protein, which includes all essential amino acids [12]. Protein hydrolysates originating from the *C. pyrenoidosa* protein have been reported to have antioxidant effects and ameliorate the development of atherosclerosis [13]. A small number of studies have targeted the emulsifier behavior of these hydrolysates. It remains unknown whether these hydrolysates are suitable emulsifiers for deriving krill O/W emulsions and whether they could contribute to inhibiting lipid oxidation. In this study, we evaluated the emulsifying, antioxidant, and lipid peroxidation inhibitory properties and the cytoprotective effects of CPHs that were hydrolyzed by alkaline proteases and neutral proteases. These two protein hydrolysates were used to emulsify krill O/W emulsions, and their physical and oxidative stabilities were investigated. The purpose of this study was to apply CPHs in a krill oil emulsion as emulsifiers, which further increased their oxidative stabilities and added to the health-adding value of the emulsion.

## 2. Results and Discussion

### 2.1. Antioxidant Activities of CPHs

The DPPH and O_2_^−^ radical scavenging activities are usually used to estimate the antioxidant activity of antioxidant components from food in vitro, since these free radicals are always found in biological systems. The effects of antioxidants on the scavenging of DPPH and O_2_^−^ probably contribute to the weakening and termination of the chain reaction of lipid oxidation. The free radicals of the DPPH scavenging activities of the neutral protease Chlorella protein hydrolysate (NCPH), the alkaline protease Chlorella protein hydrolysate (ACPH), and ascorbic acid are shown in Figure 1A. A total of 1.25 mg/mL and 2.5 mg/mL NCPH exhibited a free radical scavenging ability of about 65% for DPPH, whereas 5, 10, and 20 mg/mL NCPH scavenged nearly 100% of DPPH radicals. Additionally, 1.25, 2.5, and 5 mg/mL ACPHs scavenged DPPH radicals in a dose-dependent manner. All of the 5, 10, and 20 mg/mL ACPHs showed 100% scavenging ability. DPPH, a steady hydrophobic free radical, was shown to be reduced and eliminated by a hydrogen atom-donating compound; while hydroxyl radicals are considered to have an important role in both hydrophilic systems and the early stages of lipid peroxidation [14]. In addition, the scavenging activity of other antioxidant emulsifiers in O/W emulsions, such as whey protein isolates, lotus seedpod proanthocyanin conjugate, and Cod (*Gadus morhua*) for the protein hydrolysates of different free radicals have been examined with DPPH [15].

The O_2_^−^ free radical is an oxidant that can generate hydroxyl radicals that can abstract a hydrogen atom from other fatty acids from emulsions and forms a hydroperoxide (primary oxidation product) [16]. The scavenging activities of different concentration of CPHs on O_2_^−^ are shown in Figure 1B. NCPHs and ACPHs similarly exhibited a dose-dependent O^2^^−^ free radical scavenging activity. We also found that in the O_2_^−^ free radical scavenging experiment, NCPHs had a significantly higher antioxidant capacity than ACPHs for most concentrations of each sample. Moreover, a lower concentration (1.25 mg/mL) of NCPHs and ACPHs only showed approximately 20–30% O_2_^−^ free radical scavenging activity, whereas the value reached 70–80% at concentrations of 10 and 20 mg/mL for NCPHs and ACPHs, respectively; suggesting that the increase in dosage had a strong influence on the O^2−^ free radical scavenging activity. Consequently, such proteins and hydrolysates can act as O_2_^−^ free radical scavengers, including squid protein hydrolysates, in which the increase in the O_2_^−^ free radical scavenging activity agreed with our results [17].

Therefore, based on the results of the O^2−^ and DPPH free radical scavenging activities, CPHs does not only act as a significant free radical scavenger for O_2_^−^ and DPPH free radical-containing food or biological systems, such as in vitro and in vivo systems, but also exhibits a good potential antioxidant capability in solvent-containing systems, such as emulsion systems.

In addition, NCPHs and ACPHs have shown OH and ABTS radical scavenging activities (Appendix A), which proved that NCPHs and ACPHs are excellent antioxidants via multiple pathways.

### 2.2. Emulsifying and Foaming Activities of CPHs

To test the antioxidant activity of CPHs in emulsion, the emulsifying and foaming properties of CPHs should be preferentially determined. The EAI and ESI results of CPHs at 1.25, 2.5, 5, 10, and 20 mg/mL are shown in Figure 2A,B. In general, EAI decreased significantly and dose dependently, as did Tween 20 (positive control), whereas ESI showed no significant change. When comparing the NCPHs and ACPHs, the NCPHs seemed to have higher EAI values than those of the ACPHs at lower concentrations (1.25, 2.5, and 5 mg/mL), but they had no significant differences at higher concentrations, suggesting that neutral proteases might be more suitable for manufacturing emulsifying system-based hydrolysates. Several other hydrolysates have also been proven to have the same tendency of a decline in their emulsifying capability at higher concentrations [18,19]. Generally, there are two processes involved in the progression of emulsification, and these processes include the disruption and deformation of lipid droplets that determine the emulsion’s distinct surface area; and the emulsifier helps to stabilize this interface. Based on these two points, the protein adsorption at the oil–water interface is a diffusion method that is more easily controlled at low protein concentrations; whereas a higher protein concentration attenuates migration by the activation of an energy barrier in diffusion-dependent situations [20], which points to a decrease in emulsifying activity. This would also explain why, at high concentrations, NCPHs and ACPHs resulted in no significant differences in EAIs (Figure 2A), because extra proteins were blocked and there was no additional emulsifying activity. Moreover, the ESI results also suggested that concentration had no influence on the emulsion stability and that the NCPHs and ACPHs retained a higher emulsion stability, without exhibiting any differences from each other.

The results of the FE and FS tests of the samples with CPHs at 1.25, 2.5, 5, 10, and 20 mg/mL are shown in Figure 2C,D. In the NCPHs, the values of FE and FS increased slightly and dose-dependently and reached their maximum values at 20 mg/mL, whereas the FE value of the ACPHs did not significantly change with increasing concentration, and the FS value of the ACPHs only significantly increased at concentrations of 2.5 and 5 mg/mL. Furthermore, the FS and FE values of the NCPHs were dramatically higher than those of the ACPHs at higher concentrations (10 and 20 mg/mL), and the tendencies were similar and comparable between these two groups in the results of EAI (Figure 2A). Both FS and FE are critical foam properties that can be mediated by small-molecular-weight surfactants, such as peptides or hydrolysates. Generally, molecules’ penetration, transportation, and reorganization are involved in foam formation, which can be enhanced by the adsorption of proteins or peptides and can trigger a rapid reduction in surface tension. In some ways, the enzymatic hydrolysis of proteins, not only strongly increases their solubility, but also enhances their foaming properties, which consequently modify three major structures, as follows: a lower molecular mass, a higher amount of the ionizable group, and the essential involvement of hydrophobic structures [21]. Our results revealed that higher amounts of hydrolysates led to a better foaming ability, which was probably due to the higher speed of adsorption of the hydrolysates at the surface, which was consistent with previous reports on the foaming capability of other protein hydrolysates [18]. In addition, the emulsifying ability of the NCPHs was better than that of the ACPHs overall, indicating that neutral proteases might be more suitable for producing *Chlorella pyrenoidosa*-derived hydrolysates with a better emulsifying ability.

### 2.3. Intracellular Antioxidant Activity of CPHs In Vitro

Intracellular ROS are generated by various cell metabolite pathways in the human body. Antioxidants can regulate antioxidant enzymes in cells or directly mitigate ROS-induced damage. SOD is one of the most important antioxidant enzymes; it is ubiquitously expressed in human cells and plays a detoxifying role against superoxide radicals [22]. SODs convert superoxide radicals into O_2_ and hydrogen peroxide, and then the hydrogen peroxide catalases to oxygen and water. Therefore, we used the MDA-MB-231 cell line to assess the effect of CPHs on cellular SOD activities and H_2_O_2_-induced oxidative stress. As shown in Figure 3A, both the NCPHs and ACPHs increased SOD activity in a dose-dependent manner. Treatment with 0.25 and 0.5 mg/mL NCPHs significantly increased SOD activity in comparison to the control group (without the hydrolysate treatment), whereas treatment with 0.125, 0.25, and 0.5 mg/mL ACPHs significantly induced SOD activity. The SOD activity induced by the ACPHs was significantly higher than that induced by the NCPHs. These results indicate that NCPHs and ACPHs show antioxidant effects through intracellular detoxification. Furthermore, the antioxidant effects of the NCPHs and ACPHs against H_2_O_2_-induced cellular damage were also investigated using a DCFH-DA kit (Figure 3B,C). The cell-permeant reagent DCFH-DA is a fluorogenic dye that can reflect the peroxyl, hydroxyl, and other ROS activity in cells. After cellular uptake, internal esterase can deacetylate DCFH-DA to a non-fluorescent compound, which is later oxidized by ROS into the fluorescent compound 2′-7′dichlorofluorescein (DCF). The results indicated that the accumulation of green fluorescence was significantly enhanced by H_2_O_2_ treatment, whereas without the CPHs or H_2_O_2_ treatment, green fluorescence was hardly detected. In comparison, pretreatment with NCPHs and ACPHs dramatically attenuated H_2_O_2_-induced green fluorescence in MDA-MB-231 cells, suggesting that H_2_O_2_-induced oxidative stress was dramatically decreased. In addition, when comparing the NCPHs and ACPHs, the NCPHs were found to have better antioxidant effects against cellular oxidative damage in the MDA-MB-231 cell line, since the intensity of green fluorescence of the NCPHs was lighter than that of the ACPHs at each concentration and seemed to vanish at 0.5 mg/mL. Additionally, the peptides from protein hydrolysates in Monkfish [23] and salmon collagen hydrolysate [24] had similar effects on SOD activity or ROS mitigation effects. In this figure, we show the SOD activities induced by NCPHs and ACPHs, which could convert harmful superoxide to hydrogen peroxide, while hydrogen peroxide could also be cleaved by NCPHs and ACPHs. These data demonstrate that the NCPHs and ACPHs exerted protective effects by inducing antioxidant defenses and reducing oxidative stress. Other peptides derived from *Chlorella* sp., such as the peptide (Leu-Asn-Gly-Asp-Val-Trp) from *Chlorella ellipsiodea,* have peroxyl radical and DPPH radical scavenging antioxidant activities in vitro, and intracellular radical scavenging activity in monkey kidney cells [25]. Another peptide (Val-Glu-Cys-Tyr-Gly-Pro-Asn-Arg-Pro-Gln-Phe) from *Chlorella vulgaris* demonstrated its antioxidant activity in vitro or and in cell-based assays [26]. Antioxidant peptides NIPP-1 (Pro-GlyTrp-Asn-Gln-Trp-Phe-Leu) and NIPP-2 (Val-Glu-Val-Leu-Pro-Pro-Ala-Glu-Leu) from microalgae *Navicula incerta* have shown their cytoprotective activities in HepG2/CYP2E1 cells [27]. In addition, the abundance of aromatic amino acids (Tyr, Trp, Met, Lys, Cys, His), hydrophobic amino acids (Leu, Val), and a particular functional group (sulfhydryl group of Cys) is thought to facilitate a higher antioxidant capacity of peptides [28]. In our study, the most abundant amino acids of NCPH and ACPH were Leu, Pro, Ala, Glu, and Asp (data not shown); of which, Leu and Pro are also found in peptides of *Chlorella ellipsiodea* and *Chlorella vulgaris* and majorly facilitate higher antioxidant activities.

### 2.4. Lipid Peroxidation Inhibition Assay

The enzymatic hydrolysates of proteins in oil-dispersion products have been confirmed to have antioxidant effects on lipid oxidation, because of their radical scavenging activity and their capability for sequestering pro-oxidative metal ions. To assess the antioxidant activity of each sample in hydrophobic systems, the effects of the NCPHs and ACPHs on lipid peroxidation were evaluated by determining the levels of POV and TRABS, and the results are shown in Figure 4. The results showed that the NCPHs and ACPHs exhibited a dose-dependent POV inhibitory effect on linoleic acid oxidation, as well as a similar tendency in the TBARS inhibition experiments; which was consistent with previous reports [29]. When comparing every two days, out of six days of incubation, the inhibitory effects of the NCPHs and ACPHs against oxidation gradually increased and reached a level approximately the same as that of ascorbic acid (the positive control); whereas the antioxidant effect of ascorbic acid started to decrease from the beginning, suggesting that these two hydrolysates might have more advantages in oil–water emulsion systems with long-term inhibition activity against lipid peroxidation. When comparing these two samples, it is worth noting that the ACPHs exhibited comparatively higher inhibitory effects than those of the NCPHs based on the level of POV, even though the free radical scavenging activities of the NCPHs were higher than those of the ACPHs; this means that the antioxidant abilities of the hydrolysates in aqueous environments and emulsifying systems were equivalent but not completely the same. Therefore, determining the antioxidant capability of NCPHs and ACPHs in an oil-in-water emulsifying system is necessary.

### 2.5. CPH-Loaded Antarctic Krill Oil Emulsion

#### 2.5.1. Particle Size and Zeta Potential of CPH-Loaded Antarctic Krill Oil Emulsions

Here, we chose CPHs (1.25, 2.5, 5, and 10 mg/mL) as the emulsifiers for the Antarctic krill oil emulsions, based on their emulsifier properties. The choice of emulsifier affects the physicochemical, sensorial, and functional properties of the emulsion produced. Particle size, polymerization, and zeta potential were used to identify the emulsion’s stability. The particle sizes of the CPH-loaded Antarctic krill oil emulsions with different concentrations are shown in Figure 5A. The 1.25, 2.5, 5, and 10 mg/mL CPHs increased the particle size of the KO emulsion in a dose-dependent manner. The 1.25 and 2.5 mg/mL NCPHs-KO emulsion and the ACPHs-KO emulsion showed similar particle sizes, close to 120 nm. The NCPHs and ACPHs (10 mg/mL) clearly increased the particle size of the KO emulsions, which were approximately 257 nm and 201 nm, respectively. The zeta potentials of the different concentrations of CPH-loaded KO emulsions were all negative, and between −30 mV and −45 mV. There were no significant differences between the different concentration groups or different protease-loaded groups.

These particle sizes of the emulsion indicated that less than 5 mg/mL CPH could form small droplets, which helped in allowing active delivery, thereby boosting the rapid absorption and release of hydrophobic bioactive ingredients such as phospholipids and omega 3 fatty acids [5]. There were significant differences between the particle sizes of the NCPH-KO emulsion and the ACPH-KO emulsion, probably due to these peptides having different numbers of hydrophobic regions produced by different enzymes. The adsorption of peptides onto droplet interfaces depends on their hydrophobic properties, mainly their surface hydrophobicity [30].

The progress of the enzymatic hydrolysis of soy proteins and whey protein can produce smaller peptides that contain a partially exposed hydrophobic core and fewer secondary and tertiary structures, making them excellent emulsifiers [31]. These properties account for their increased dispersion in the oil–water interface and the maintenance of the physical and chemical stability of emulsions [31]. However, sometimes, a protein hydrolysate alone, such as porcine bone protein hydrolysates, cannot obtain the stabilities achieved in O/W emulsions [6].

#### 2.5.2. Morphology of CPH-Loaded KO Emulsions

The different concentrations of the CPH-loaded KO emulsions showed that both the NCP- and ACP-loaded KO emulsions were homogenous, opaque, and stable on day 0. With the addition of CPHs, the initial reddish colloids turned brownish, which possibly occurred because the color of the CPHs changed the color of the emulsions (Figure 5C). During the 30-day storage time, the 1.25, 2.5, and 10 mg/mL NCPH- and ACPH-loaded KO emulsions had different degrees of creaming or separation from the first three days. The 10 mg/mL NCP- and ACP-loaded KO emulsions had the most obvious signs of separation (data not shown). However, the 5 mg/mL NCPH- and ACPH-loaded KO emulsions remained stable at room temperature for 30 days (Figure 5D). These results may be because an adequate surface hydrophobicity of CPHs is needed for strong and cohesive films to form around droplets, and 1.25 mg/mL and 2.5 mg/mL were not sufficient for the CPHs to function as good emulsifiers. The exceeded protein load at the interface led to extra protein escaping from the interface, to form flocculation via a hydrophobic interaction, making the 10 mg/mL CPH-loaded emulsion easy to cream and separate.

The microstructure of the CPH-loaded KO emulsions with different concentrations was observed using CSLM. Using the CSLM image, the protein (which exhibited green fluorescence) and oil droplet (which exhibited red fluorescence) distributions were obtained. As shown in Figure 5E, the 1.25, 2.5, and 5 mg/mL NCPH-loaded KO emulsions and ACPH-loaded KO emulsions had a relatively uniform distribution of oil droplets, with small particles and good dispersions. The 10 mg/mL NCPH-loaded KO emulsions and ACPH-loaded KO emulsions had larger particles and more protein in the oil droplets. These results were consistent with the particle results in Figure 5A and indicate that NCPHs and ACPHs can be applied in KO emulsions as emulsifiers.

### 2.6. Oxidative Stability of the Emulsion

The enzymatic hydrolysate of protein in oil-dispersion products has been reported to suppress lipid oxidation, due to their peptide radical scavenging and pro-oxidative metal ion sequestering capability. The NCPHs and ACPHs showed antioxidant activities against KO oxidation (Figure 1 and Figure 4). However, the oxidative stability of the NCPH- and ACPH-loaded emulsions remains unknown. Here, we used the levels of POV and TBARS to represent the oxidative stability. The KO emulsion and Tween 20 emulsion were used as controls. After different the KO emulsions were stored at 40 °C for six days, the POV value of the KO emulsion and Tween 20 emulsion increased from 3.43 meq/kg to 3.77 meq/kg, and 2.56 meq/kg to 3.62 meq/kg, respectively. However, the POV value of the NCPH-loaded emulsion increased from 1.21 meq/kg to 1.59 meq/kg, and the POV value of the ACPH-loaded emulsion increased from 1.01 meq/kg to 1.19 meq/kg. These results indicate that the POVs of the NCPH- and ACPH-loaded emulsions were both significantly lower than those of the KO emulsions, which was probably due to the lipid oxidation inhibitory effect. After six days of storage at 40 °C, the TRABS values of the KO emulsion and the Tween 20-, NCPH-, and ACPH-loaded KO emulsions were significantly increased and finally reached 25.65 mg/kg, 19.04 mg/kg, 12.76 mg/kg, and 9.58 mg/kg, respectively (Figure 6B). These results indicate that Tween 20, NCPHs, and ACPHs significantly inhibited the TRABS formation in the KO emulsions.

We hypothesize that the Tween 20-loaded KO emulsions decreased the POV and TRABS values, probably through physical effects, because Tween 20 can form interfacial layers that may sterically hamper the movement of pro-oxidants from the water to oil phases, thereby improving the oxidative stability. The change in POV and TRABS values in the NCPH- and ACPH-derived emulsions was more significant than in the Tween 20-loaded KO emulsions, which was possibly because of the antioxidant effect via the free radical scavenging activities of the NCPHs and ACPHs; this result agrees with the finding that hydrolysates in emulsions such as fish protein hydrolysates, rice dreg protein hydrolysate [8], and porcine bone protein hydrolysates [6] have antioxidant effects in water/oil emulsions. It has also been reported that antioxidative proteins can inhibit emulsion lipid oxidation through the interface effect and continuous phase effect during storage [32]. The interface of a protein-loaded O/W emulsion always has a higher degree of rigidity and continuity, which would likely prevent the penetration and diffusion of radicals into the lipid phase within the emulsion droplets [33]. Antioxidant protein hydrolysates in the continuous phase were thought to break the free radical chain and prevent the formation of peroxides by reacting with certain precursors of peroxide.

## 3. Materials and Methods

### 3.1. Chemicals

Commercial *Chlorella Pyrenoidosa* powder and Antarctic krill oil were obtained from Dalian Jianyang Biological Company (Dalian, China) and Liaoyu Group Co., Ltd. (Dalian, China), respectively. The neutral proteases (EINECS:253-457-5) and alkaline proteases (EINECS:232-752-2) were obtained from Yuanye Biotechnology Co., Ltd. (Shanghai, China). Ferrous chloride and ascorbic acid were obtained from Damao Chemical Reagent Factory (Tianjin, China). Sodium dodecyl sulfate (SDS), linoleic acid (stored at −20 °C), 1,1,3,3-tetraethoxypropane, trichloroacetic acid, Thiobarbituric acid (TBA), Nile Red, and Nile Blue were bought from Aladdin Industrial Co., Ltd. (Shanghai, China). Ammonium thiocyanate was bought from Sinopharm Group Reagent (Shanghai, China). Trypsin-EDTA, fetal bovine serum (FBS), and DMEM cell culture medium were bought from Invitrogen (Carlsbad, CA, USA). A superoxide dismutase (SOD) assay kit and reactive oxygen species (ROS) assay kit were bought from Nanjing Jiancheng Bioengineering Research Institute (Nanking, China).

### 3.2. Hydrolysates from Chlorella Protein

A total of 20 g Chlorella powder was dissolved in 2 L purified water (power: solution (*w/v*) = 1:100) to prepare a Chlorella powder solution. Before hydrolysis, the solution pH was adjusted to 7.5 (neutral proteases) and 9.0 (alkaline proteases), and the solution temperature was set at 45 °C (neutral proteases) and 50 °C (alkaline protease). Then, 20,000 U neutral protease and 20,000 U alkaline protease were added to the Chlorella powder solution for 5 h. Then, this process was stopped by performing incubation in boiling water for 10 min. The hydrolysate was observed by precipitation with 95% ethanol (hydrolysate: 95% ethanol (*v/v*) = 1:4), and the supernatant of the hydrolysates was then freeze-dried.

### 3.3. Determination of the Antioxidant Activity of Chlorella Protein Hydrolysates (CPHs)

The DPPH, O_2_^−^, HO·, and ABTS scavenging activities were measured, to represent the antioxidant activities of the CPHs and ascorbic acid (positive control) at 1.25, 2.5, 5, 10, and 20 mg/mL. The methods we used were in accordance with those of other reports, with small modifications [19,34,35,36].

#### 3.3.1. DPPH Radical Scavenging Activity Assay

A total of 2 mL of CPHs or ascorbic acid was blended with 2 mL of DPPH (0.04 mg/mL) and kept in the dark for 30 min. Then, these solutions were centrifuged at 5000 r/min for 10 min; then, we measured the absorbance at 517 nm [34]. The calculation formula was as follows:(1)$$\mathrm{DPPH}\;\mathrm{radical}\;\mathrm{scavenging}\;\mathrm{activity}\;\left(\%\right)=\left(1-\frac{\left(A_{x}{-A}_{x0}\right)}{A_{0}}\right)\times 100\%$$
where $A_{x}$ is the value of sample absorbance; $A_{x0}$ is the value of the interference group absorbance (distilled water instead of DPPH); and A_0_ is the absorbance of distilled water instead of the sample group.

#### 3.3.2. Hydroxyl Radical (HO) Scavenging Activity

A total of 1 mL of CPHs or ascorbic acid was blended with 1 mL of salicylic acid at 9 mM, 1 mL of FeSO_4_ at 9 mM, and 1 mL of H_2_O_2_ at 8.8 mM and was incubated at 37 °C for 30 min. Then, the absorbances of these mixtures were tested at 510 nm [36]. The calculation formula was as follows:(2)$$\mathrm{HO}\;\mathrm{radical}\;\mathrm{scavenging}\;\mathrm{activity}\;(\%)=\left(1-\frac{\left(A_{x}{-A}_{x0}\right)}{A_{0}}\right)\times 100\%$$
where $A_{x}$ is the value of sample absorbance; $A_{x0\;}$ is the value of the interference group absorbance (distilled water instead of H_2_O_2_); and A_0_ is the absorbance of distilled water instead of the sample group.

#### 3.3.3. Superoxide Anion Radical (O_2_^−^) Scavenging Activity

A total of 1 mL of CPHs or ascorbic acid was prepared and mixed with 1 mL of Tris-HCl at 50 mM and 0.6 mL of pyrogallol at 25 mM, and was incubated at 25 °C for 5 min. The reaction was stopped using HCl, and its absorbance was read at 299 nm [35]. The calculation formula was as follows:(3)$$O^{2-}\;\mathrm{radical}\;\mathrm{scavenging}\;\mathrm{activity}\;(\%)=\left(1-\left(A_{x}-\frac{A_{x0}}{A_{0}}\right)\right)\times 100\%$$
where $A_{x}$ is the value of sample absorbance; $A_{x0}$ is the value of the interference group absorbance (distilled water instead of pyrogallol); and A_0_ is the absorbance of distilled water instead of the sample group.

#### 3.3.4. Determination of ABTS Scavenging Activity

The ABTS solutions were prepared with 7.4 mM ABTS and an equal volume of 2.45 mM potassium persulfate, kept in the dark for 15 h and diluted with PBS (pH 7.4) until the absorbance at 734 nm reached 0.70 ± 0.02. The 10 μL of CPHs, together with the 90 μL diluted ABTS solution, were incubated for 6 min in the dark and measured at 734 nm [19]. The calculation formula was as follows:(4)$$\mathrm{ABTS}\;\mathrm{radical}\;\mathrm{scavenging}\;\mathrm{activity}\;(\%)=\left(1-\frac{\left(A_{x}{-A}_{x0}\right)}{A_{0}}\right)\times 100\%$$
where $A_{x}$ is the value of sample absorbance; $A_{x0}\;$ is the value of the interference group absorbance (distilled water instead of ABTS); and A_0_ is the absorbance of distilled water instead of the sample group.

### 3.4. Emulsifying Properties

The emulsification activity and the emulsion stability of CPHs on Antarctic krill oil were measured according to Pearce and Kinsella’s method with modifications [37]. Each of 8 mL of 1.25 mg/mL, 2.5 mg/mL, 5 mg/mL, 10 mg/mL, and 20 mg/mL CPHs–distilled water solutions and 2 mL of Antarctic krill oil were mixed together and homogenized for 10 min at 13,500 rpm (T 10 basic ULTRA-TURRAX, IKA, Freiburg, Germany). Before and after the 10 min homogenization, 50 μL of the mixture sample from the middle layer of the emulsion was taken. Each sample was diluted 100-fold with 0.1% sodium dodecyl sulfate and shaken for 10 s. The absorbance of the sample was measured at 500 nm. The calculation formula was as follows:(5)$${\mathrm{EAI}\;(m}^{2}/g)=\frac{(2\;\times \;2{.303\;\times \;A\;\times \;D}_{F})}{l\emptyset C}$$
where A = A_500_, D_F_ = dilution factor (100), l = passage path length of the cuvette (m), ∅ = oil fraction, and C = protein concentration in the aqueous phase (g/m^3^);
(6)$$\mathrm{ESI}\;\left(\min\right)=\frac{A_{0\;}\times \;\Delta t}{\Delta A}$$
where Δt = 10 min and $\Delta A=A_{0}-A_{10}$, $A_{0}$, and $A_{10}$ are the absorbance of the sample at 0 min and 10 min after homogenization, respectively.

### 3.5. Foaming Properties

Foam expansion (FE) and foam stability (FS) were determined according to the method described by Shahidi and Synowiecki, with some adjustments [38]. Ten milliliters of protein hydrolysate solution (1.25 mg/mL, 2.5 mg/mL, 5 mg/mL, 10 mg/mL, and 20 mg/mL) were homogenized at 13,500 rpm for 1 min at room temperature. The solutions needed to be at standard conditions, and their volumes were recorded at 0 min and 10 min. The calculation formula was as follows:(7)$$\mathrm{FE}\;\left(\%\right)=\left(\frac{V_{T}}{V_{o}}\right)\times 100\;\mathrm{FS}\;\left(\%\right)=\left(V_{t}-V_{o}\right)/\left(V_{T}-V_{o}\right)$$
where $V_{T}$ is the total volume after homogenization; $V_{o}$ is the original volume before homogenization; and $V_{t}$ is the volume after standing at room temperature for 10 min.

### 3.6. Activity of SOD

MDA-MB-231 cells (1 × 10^6^) from the American Type Culture Collection were cultured for 24 h and treated with CPHs to final concentrations of 0, 0.125, and 0.25, 0.5 mg/mL for 3 h. The cells were scraped and centrifuged at 1500 rpm/min (5 min) for collection. These cell pellets were washed and sonicated at 300 W to obtain the cell lysates. The SOD activity of the cell lysate was measured using a SOD kit, as stated in the manufacturer’s manual (Nanjing Jiancheng Bioengineering Research Institute, Nanjing, China). The total protein content in the samples was measured using a BCA protein assay kit.

### 3.7. Reactive Oxygen Species (ROS) Level Determination

The ROS level was measured using an ROS kit, as stated in the manufacturer’s manual (Nanjing Jiancheng Bioengineering Research Institute, Nanjing, China). MDA-MB-231 cells were incubated in DMEM culture medium containing 10% FBS for 24 h and then pretreated with CPHs to final concentrations of 0, 0.125, 0.25, and 0.5 mg/mL for 3 h. Afterward, the entire cell medium was freshly replaced. H_2_O_2_ was added to the cells and incubated for 3 h at a final concentration of 300 µM. Then, 100 µL of 2′-7′dichlorofluorescin diacetate (DCFH-DA,10 mM) was incubated with these cells for 30 min. The fluorescence of total ROS (T-ROS) was measured using fluorescence microscopy (Olympus IX81, Tokyo, Japan) (excitation wavelength: 485 nm, emission wavelength: 535 nm).

### 3.8. Lipid Peroxidation Inhibitory Activity Assay of CPHs

A total of 0.1 mL of linoleic acid together with 10 mL of 99.5% ethanol solution were added to CPHs (0, 1.25, 2.5, 10, 20 mg/mL) or 10 mg/mL ascorbic acid as a positive control in 10 mL of 0.2 M PBS (pH 7.2). The mixed solutions were placed in a 40 °C light-proof room for 0, 2, 4, or 6 days. TBA values were measured using the method of Ohkawa et al. [39]. The POV value was measured according to the Chinese National Standard GB 5009.227-2016 [40]. The calculation formula was as follows:(8)$$\mathrm{Inhibition}\;(\%)=[\frac{A_{0}-A_{1}}{A_{0}}]\;\times \;100$$
where A_0_ is the absorbance of the control reaction (the addition of the hydrolysate is 0 mg) on the sixth day, and A_1_ is the absorbance in the presence of the sample on that day.

### 3.9. Emulsion Preparation

A total of 1 mL of KO together with 9 mL of CPHs (0, 1.25, 2.5, 5, 10 mg/mL) or tween 20 (5 mg/mL) in distilled water were firstly dispersed at 13,500 rpm for 7 min using an IKA disperser (IKA^®^ Works, Inc., Wilmington, NC, USA) to form primary emulsions. These emulsions then underwent 20 min ultrasonication at 658 W in an ultrasonic crusher (SCIENTZ-IID, Ningbo, China), to form the final emulsions

### 3.10. Particle Size and Zeta Potential Measurements

Both particle sizes and zeta potentials were measured using a dynamic light scattering Zetasizer (Brookhaven Instruments Corporation, Holtsville, NY, USA) to reflect the physical condition of the emulsions. The emulsion samples were diluted 50 times and 1000 times with hyperpure water before the particle size and zeta potential measurements, to obtain a suitable intensity. The experiment was performed at room temperature in triplicates.

### 3.11. Confocal Laser Scanning Microscopy (CLSM)

CLSM was applied to reflect the microstructures of the CPH emulsions. The CLSM samples were prepared with different emulsions (1 mL) and 40 µL of Nile Red (0.1%, *w/v*) and Nile Blue (0.1%, *w/v*). A total of 10 µL of the CLSM sample was fixed on slides using nail polish and examined with a 100× magnification lens (excitation at 488 nm, excitation at 633 nm).

### 3.12. Oxidative Stability of CPH Emulsions

The oxidative stability of the CPH emulsions, the KO emulsions, and the Tween 20 emulsions was expressed by POVs and TRABS in an accelerated oxidation reaction (40 °C) for 6 days. The oil from their emulsions was extracted using a previously described hexane extraction method [41]. The POV and TRABS of the emulsions were sampled every 2 days for analysis. The POV value was measured. The PVs were determined as stated in the method of Undeland et al. [42]. TRABS was determined based on the method of the Chinese National Standard GB 5009, 181–2016 [43].

### 3.13. Statistical Analysis

The statistical analysis of each group was performed using SPSS 22 software (SPSS Inc., Chicago, IL, USA). One-way ANOVA with Tukey’s test was used to analyze the homogeneity of variance, where *p* < 0.05 was considered significant. All the experiments were performed three parallel times, and the data are expressed as the SD ± means.

## 4. Conclusions

The present study showed that neutral protease protein hydrolysates (NCPHs) and alkaline protease protein hydrolysates (ACPHs) from *Chlorella pyrenoidosa* have antioxidant, emulsifying, and foaming activities. The NCPHs and ACPHs also induce SOD activities and mitigate H_2_O_2_-induced ROS levels in vitro. In addition, the NCPHs or ACPHs inhibited linoleic acid oxidation, and the NCPH- and ACPH-derived krill O/W emulsion remained physically stable for at least one month, where their POV and TRABS were lower than those of the KO and Tween 20-derived emulsions. Our study suggests that NCPHs and ACPHs have the potential to be applied in krill oil-in-water emulsions, as both emulsifiers and antioxidants.

## Figures and Tables

**Figure 1 marinedrugs-20-00345-f001:**
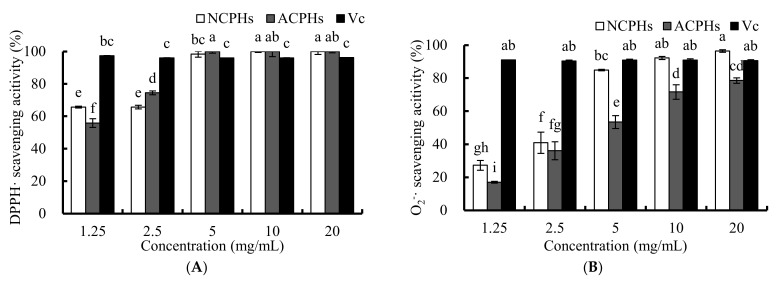
DPPH (**A**) and O_2_^−^ (**B**) free radical scavenging activities of 0, 1.25, 2.5, 5, 10, and 20 mg/mL CPHs and VC. CPHs were obtained by neutral proteases (20,000 U) and alkaline proteases (20,000 U) for 5 h, separately. Different letters above values indicate different significances (*p* < 0.05).

**Figure 2 marinedrugs-20-00345-f002:**
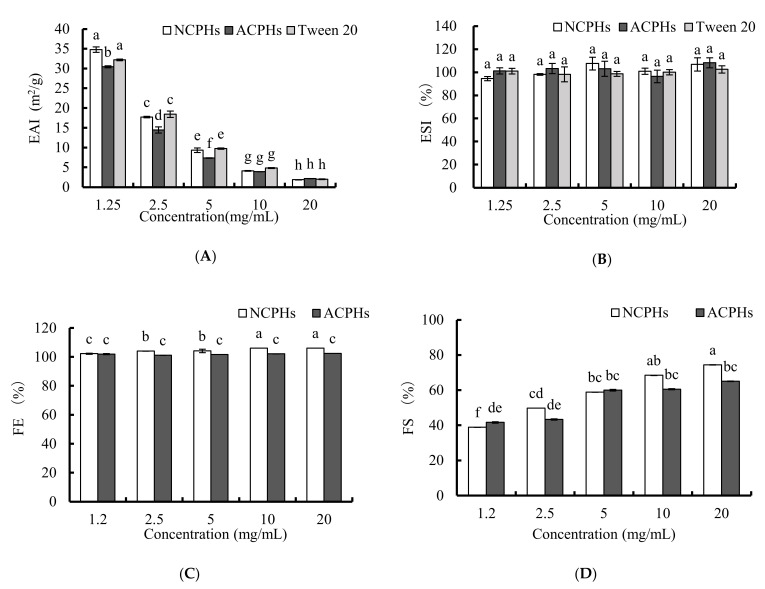
EAI (**A**) and ESI (**B**) of NCPHs, ACHPs, and Tween 20 at 0, 1.25, 2.5, 5, 10, and 20 mg/mL. FE (**C**) and FS (**D**) of NCPHs and ACHPs at 0, 1.25, 2.5, 5, 10, and 20 mg/mL. Different letters (a–h) above values indicate different significances (*p* < 0.05).

**Figure 3 marinedrugs-20-00345-f003:**
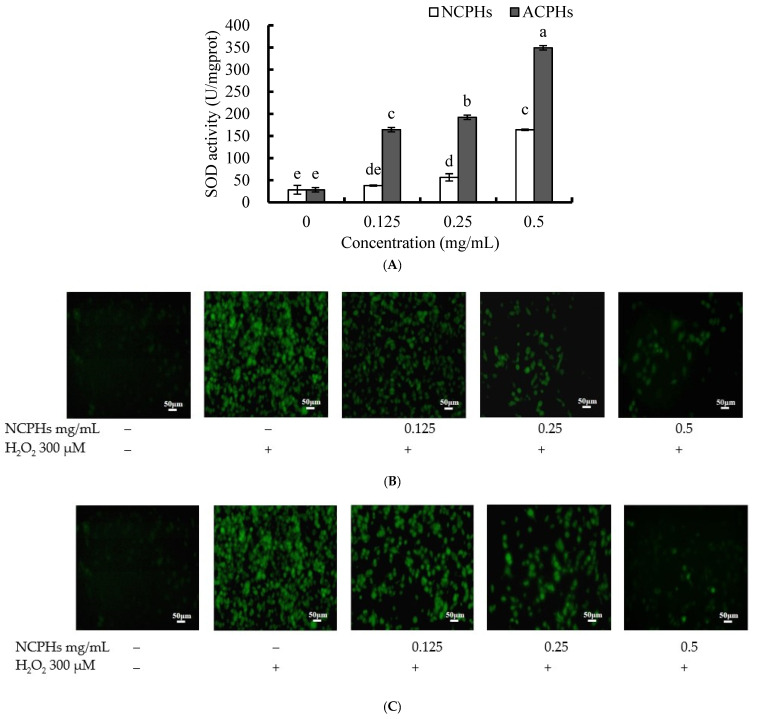
A total of 1 × 10^6^ MDA-MB-231 cells were treated with 0, 0.125, 0.25, and 0.5 mg/mL NCPHs and ACPHs for 3 h. They were then pretreated with 0 or 300 μM H_2_O_2_ for another 3 h. (**A**) SOD activities of MDA-MB-231 cells after NCPH and ACPH treatments. (**B**) ROS level of MDA-MB-231 cells after treatment with H_2_O_2_ or NCPHs. (**C**) ROS level of MDA-MB-231 cells after treatment with H_2_O_2_ or ACPHs. Different letters (a–e) above values indicate different significances (*p* < 0.05).

**Figure 4 marinedrugs-20-00345-f004:**
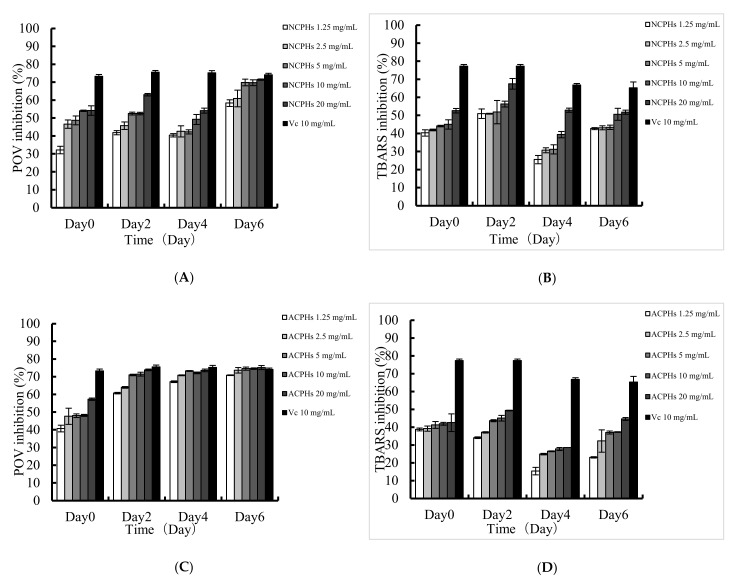
Effect of different concentrations of NCPHs and ACPHs on linoleic acid oxidation were measured at 40 °C for 6 days. Lipid peroxidation inhibition assay of CPHs: POV inhibition of NCPHs (**A**) and ACPHs (**C**) in concentrations of 1.25, 2.5, 5, 10, and 20 mg/mL for 0, 2, 4, and 6 days. TRABS inhibition of NCPHs (**B**) and ACPHs (**D**) in concentrations of 1.25, 2.5, 5, 10, and 20 mg/mL for 0, 2, 4, and 6 days.

**Figure 5 marinedrugs-20-00345-f005:**
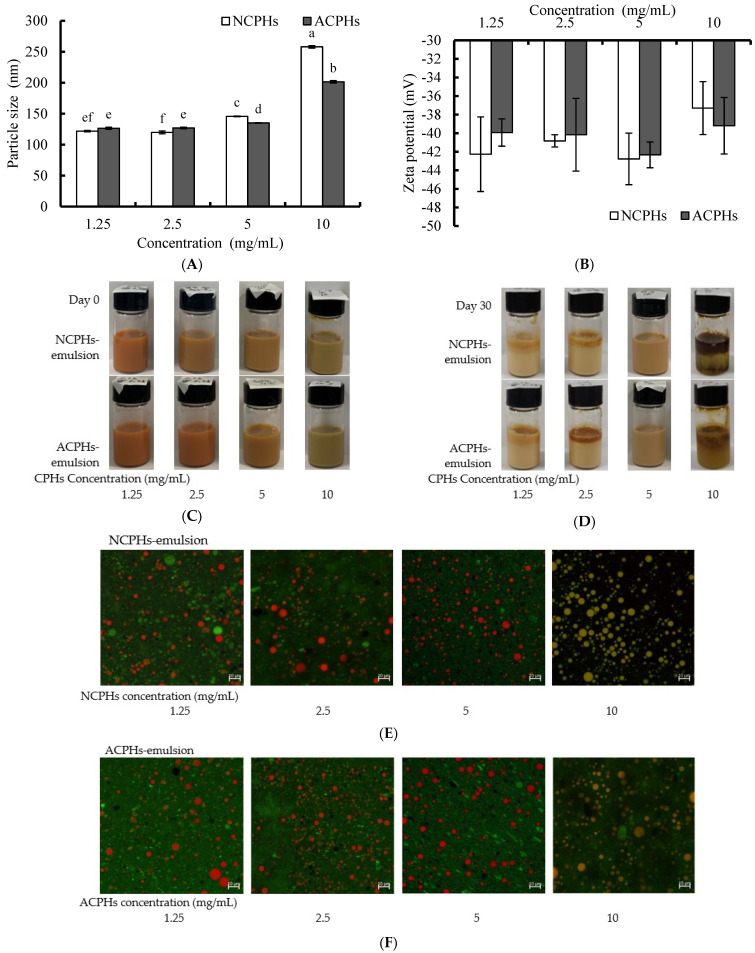
NCPHs and ACPHs in concentrations of 1.25, 2.5, 5, 10, and 20 mg/mL in water were used to form a KO emulsion containing 10% krill oil. (**A**) Particle size of CPH-derived KO emulsions. (**B**) Zeta potential of CPH-derived KO emulsions. (**C**) Photograph of CPH-derived KO emulsions at day 0. (**D**) Photograph of CPH-derived KO emulsions at day 30. (**E**) CLSM micrographs of NCPH-derived KO emulsions. (**F**) CLSM micrographs of ACPH-derived KO emulsions. Different letters (a–f) above values indicate different significances (*p* < 0.05).

**Figure 6 marinedrugs-20-00345-f006:**
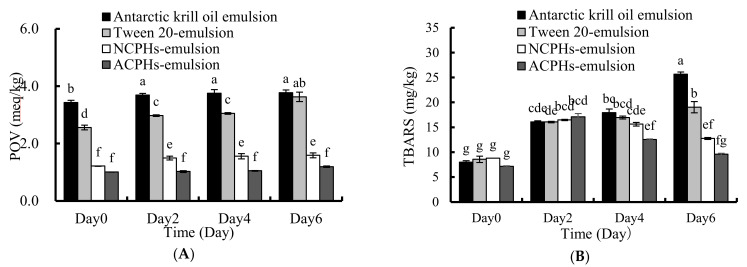
Accelerated oxidation of CPH-derived emulsion, KO emulsions, and Tween 20 KO emulsions at 40 °C for 6 days. Lipid peroxidation value: (**A**) POV value of KO emulsion, Tween 20 emulsion, and CPH-derived emulsion over 6 days. (**B**) TRABS value of KO emulsion, Tween 20 emulsion, and CPHs derived emulsion over 6 days. The significance was analyzed by one-way ANOVA followed by Tukey’s HSD using SPSS 16.0. The different letters (a–g) above the bars indicate significant differences among the treatments for each condition (*p* < 0.05).

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
