# Peer review of "Antioxidative Effect of Chlorella Pyrenoidosa Protein Hydrolysates and Their Application in Krill Oil-in-Water Emulsions"

_marinedrugs, 2022, doi:10.3390/md20060345_

Round 1

Reviewer 1 Report

The authors of “Antioxidative Effect of Chlorella Pyrenoidosa Protein Hydrolysates and Their Application in Krill Oil-In-Water Emulsions” presents a analytical analysis of the potential use of microalgae protein hydrolysates. However, the present version of the manuscript (MS) present relevant weakness that needs major revision.

General comments:

The text English style must be revised. Not only for the typographical and/or grammatical errors, the text must avoid curly style , unexpected explanations and focus in the specific results that support your main hypothesis. There are too much irrelevant data in the present version of the MS that hide the relevant ones.

Abbreviations must be explained in its first appearance, review the whole paper to avoid it (Example, line 24)

Avoid material and methods in the abstract.

Figure legends must bring enough information to be independently understandable without read the main text. All the figure legends must be deeply rewritten in this sense.

Specific comments:

Results and discussion:

The text has been structured following the experimental protocols (line 72). In my opinion, the relevant is not the specific assay, it must be the aim of the experiment (antioxidant activity). It will improve the quality of the MS.

Graphic and figures: There are a lot of data that seems crude data and poor explicative. For example, figure 1 shows minor differences. In my opinion, this data must be in supplementary material and include in the text only the relevant ones. This statements may be applied to all MS figures. The main aim must be highlight the relevant data.

Material and methods:

Line 342: Did you use Chlorella powder? Commercial? Lyophilized? Did you use replicates of different batch? The used biological material needs a deep explanation.

Line 344: What exactly means neutral or alkaline proteases? Do you have EC Numbers?

Line 355: Why 20g/2L?

Line 361: Normally, protein precipitation is performed with acetone 100%, did you have references or support for ethanol application ?

Page 12 Lines 365-404: The references of each method must be included.

Line 433: check the data

Conclusion section must be rewritten accordantly

Author Response

Response to reviewers’ comments

Reviewer #1:

  1. The authors of “Antioxidative Effect of Chlorella Pyrenoidosa Protein Hydrolysates and Their Application in Krill Oil-In-Water Emulsions” presents a analytical analysis of the potential use of microalgae protein hydrolysates. However, the present version of the manuscript (MS) present relevant weakness that needs major revision.

Response: Thank you for the reviewers for their careful reading of the manuscript and constructive suggestions. We have revised the manuscript according to the suggestions of the reviewers. Revised portions have been underlined in red on the revised manuscript. We believe that the paper is improved and hope that the corrections made together with the attached reply satisfy the concerns raised.

  1. The text English style must be revised. Not only for the typographical and/or grammatical errors, the text must avoid curly style , unexpected explanations and focus in the specific results that support your main hypothesis. There are too much irrelevant data in the present version of the MS that hide the relevant ones.

Response: According to the reviewer’s suggestion, we have revised English style of this paper. Some typographical and/or grammatical errors are corrected. In particular, some data such as O2- and ABTS free radical scavenging activity of CPHs were translated to supplementary data, the irrelevant explanation was deleted. (Line: 116-118, Page: 3)

  1. Abbreviations must be explained in its first appearance, review the whole paper to avoid it (Example, line 24)

Response: According to the reviewer’s suggestion, abbreviations of POV, TRABS, SOD, ROS have been explained separately. (Line: 24-27, Page: 1)

  1. Avoid material and methods in the abstract.

Response: According to the reviewer’s suggestion, some description was changed to avoid material and methods. The older one: it was used to prepare protein hydrolysates through neutral proteases and alkaline proteases. Here, Chlorella protein hydrolysate (CPH) (1.25, 2.5, 5, 10, 20 mg/mL) was evaluated for its antioxidant and emulsifying activities and its effects on krill oil-in-water (O/W) emulsions. The new sentence: we assessed the antioxidant and emulsifying effects of Chlorella protein hydrolysate (CPH) through neutral proteases and alkaline proteases, and the properties’ of CPHs derived krill oil-in-water (O/W) emulsions. (Line: 15-18, Page: 1)

  1. Figure legends must bring enough information to be independently understandable without read the main text. All the figure legends must be deeply rewritten in this sense.

Response: According to the reviewer’s suggestion, all figure legends were rewritten in details. (Line: 121-123, Page: 3; Line: 170-172, Page: 5; Line: 223-227, Page: 6; Line: 253-257, Page: 8; Line: 290-294, Page: 9-10; Line: 356-359, Page: 11)

  1. The text has been structured following the experimental protocols (line 72). In my opinion, the relevant is not the specific assay, it must be the aim of the experiment (antioxidant activity). It will improve the quality of the MS.

Response: Here only DPPH and O2- scavenging activities were present in Fig. 1 which because DPPH is a stable free radical and accepts an electron or hydrogen to become a stable molecule, and superoxide radical is very important in the initial reaction and the chain reaction of lipid oxidation. The O2- free radical is an oxidant that can generate hydroxyl radicals that can abstract a hydrogen atom from other fatty acid from emulsions and forms a hydroperoxide (primary oxidation product). The enhancement of DPPH radical scavenging activity suggested that hydroxyl groups of CPHs play a key role in donating hydrogen and electrons, which lead to the termination of the radical chain reaction. And the termination of the radical chain reaction in emulsion will help keep the oxidative stabilities of O/W emulsions. (Line:82-84; 86-90; 93-97, Page 2; Line:98-101, Page: 3)

  1. Graphic and figures: There are a lot of data that seems crude data and poor explicative. For example, figure 1 shows minor differences. In my opinion, this data must be in supplementary material and include in the text only the relevant ones. This statements may be applied to all MS figures. The main aim must be highlight the relevant data.

Response: According to the reviewer’s suggestion, we majorly revised the result and discussion part in the Fig 1 of this text. After revising, only DPPH and O2- scavenging activities were shown in Fig. 1 because there was relevant proof that the scavenging of DPPH and O2- of antioxidant emulsifier were important in oil or O/W emulsion system. In addition, we explained the data again and add new discussion and their reference. The OH and ABTS scavenging activities were shown in supplementary data. (Line 116-120, Page: 3)

  1. Line 342: Did you use Chlorella powder? Commercial? Lyophilized? Did you use replicates of different batch? The used biological material needs a deep explanation.

Response: Thank you for asking. Here we used commercial Chlorella Pyrenoidosa powder from Dalian Jianyang Biological Company (Dalian, China) which this information was added. And we used different batch of Chlorella powder in this experiment. (Line 362, Page:11)

  1. Line 344: What exactly means neutral or alkaline proteases? Do you have EC Numbers?

Response: Thank you for asking. Neutral protease (also named Dispase) is a kind of endoprotease that is deeply fermented from selected 1398 Bacillus subtilis and refined using advanced techniques. In certain temperature and pH environment, it can decompose macromolecule proteins into polypeptide and amino acid products, and transform into unique hydrolyzed flavors. It can be used in the field of protein hydrolysis, such as food, feed, cosmetics, and nutrition areas. Alkaline protease (also named alcalase protease) is a protein enzyme, it comes from Bacillus licheniformis after its fermentation and refinement, it mainly composed by Bacillus licheniformis protease. It's an endoprotease of serine, can hydrolysis the macro-molecule protein into free amino acid etc. Here we added EC numbers of neutral or alkaline proteases in this manuscript. Neutral protease (EINECS:253-457-5) and alkaline protease (EINECS:232-752-2). (Line: 364-365; Page:11)

  1. Line 355: Why 20g/2L?

Response: Thank you for asking. Here we dissolved 20 g Chlorella powder in 2 L purified water because we settled power: solution (w/v) to 1: 100 which same condition was also used by Zixu Wang (Wang Z, Liu X, Xie H, et al. Antioxidant activity and functional properties of Alcalase-hydrolyzed scallop protein hydrolysate and its role in the inhibition of cytotoxicity in vitro[J]. Food Chemistry, 2021, 344: 128566.)

  1. Line 361: Normally, protein precipitation is performed with acetone 100%, did you have references or support for ethanol application ?

Response: Thank you for asking. Here we revised these sentence in details which the hydrolysate was observed by precipitation with 95% ethanol (hydrolysate: 95% ethanol (v/v) = 1: 4) and supernatant of hydrolysates was then freeze-dried. This method was also used by used by Zixu Wang (Wang Z, Liu X, Xie H, et al. Antioxidant activity and functional properties of Alcalase-hydrolyzed scallop protein hydrolysate and its role in the inhibition of cytotoxicity in vitro[J]. Food Chemistry, 2021, 344: 128566.) (Line:383, Page: 12)

  1. Page 12 Lines 365-404: The references of each method must be included.

Response: Thank you for your suggestion. And each method was followed by references. (Line:389,393, 402, 411, Page:12)

  1. Line 433: check the data

Response: Thank you for reminding. The cells were scraped and centrifuged at 1500 rpm/min (5 min) for collection. (Line: 457, Page: 13)

  1. Conclusion section must be rewritten accordantly

Response: Thank you for your suggestion. And we rewritten the conclusion. (Line 514-522, Page:15)

Reviewer 2 Report

Well done, nice work.

!

Author Response

Response to reviewers’ comments

Reviewer #2:

Well done, nice work.

!

Response: Thank you for the reviewers for their careful reading of the manuscript and constructive suggestions. We have revised the manuscript according to the suggestions of the reviewers. Revised portions have been underlined in red on the revised manuscript. We have revised English style of this paper. Some typographical and/or grammatical errors are corrected.

Reviewer 3 Report

Herein, the authors aimed to apply CPHs in a krill oil emulsion as emulsifiers, which further increased their oxidative stabilities and added to the health-adding value of the emulsion; the study is good but certain points should be carefully addressed

Major point

  • The structures of the major metabolites in the Chlorella pyrenoidosa protein hydrolysates should be identified and consequently drawn and correlated with the activity

Minor points

  • What is new should be highlighted more comprehensively in the introduction section
  • There are too many references that make it similar to a review not an original manuscript
  • There are some spelling and grammatical that should be revised carefully

Author Response

Response to reviewers’ comments

Reviewer #3:

  1. Herein, the authors aimed to apply CPHs in a krill oil emulsion as emulsifiers, which further increased their oxidative stabilities and added to the health-adding value of the emulsion; the study is good but certain points should be carefully addressed.

Response: Thank you for the reviewers for their careful reading of the manuscript and constructive suggestions. We have revised the manuscript according to the suggestions of the reviewers. Revised portions have been underlined in red on the revised manuscript. We believe that the paper is improved and hope that the corrections made together with the attached reply satisfy the concerns raised.

Major point

  1. The structures of the major metabolites in the Chlorella pyrenoidosaprotein hydrolysates should be identified and consequently drawn and correlated with the activity

Response: Thanks for the comments. We agreed this suggestion by reviewer. In this study, a series of antioxidant capacity of NCPH and ACPH, including in aqueous phase and emulsion system, were investigated. In order to illustrate the structure-function relationship, the amino acid sequence of the NCPH and ACPH were planned to be identified by the Shanghai Omicsolution Co.,Ltd. (Beijing, China). We had sent the samples twice to this company within this month, unfortunately, this company was quarantined by administration because of the pandemic outbreak of COVID-19 in Beijing. The entire team of workers were told to stay at home till the next instructions that nobody knows when it will be issued. As a matter of fact, not only Beijing but also the other major cities (i.e. Shanghai) where such kind of company exists are all in the influence of COVID-19. Therefore, we think the identification of amino acid sequences of NCPH and ACPH could not be accomplished within this short period. We are sorry for the troubles. However, as suggested, we will search for more papers and write more discussions of structure-function relationship in this manuscript. (Line: 207-220, Page: 5,6). In addition, we have planned to another paper which specially and deeply investigated structure of NCPH and ACHP which have emulsified activities containing the amino acid sequence results.

Minor points

  1. What is new should be highlighted more comprehensively in the introduction section

Response: Thank you for asking. And we add new contents in introduction part where we emphasize the oxidative problem of O/W emulsion which is all suit of krill O/W emulsion. And if CPHs are proper emulsifiers to derive krill O/W emulsion and contribute to inhibit lipid oxidation still remain unknown. (Line:47-50, 59, 70-71, Page:2)

  1. There are too many references that make it similar to a review not an original manuscript

Response: Thank you for your suggestions, and we deleted some not necessary references in this manuscript.

  1. There are some spelling and grammatical that should be revised carefully

Response: Thank you for your suggestion, we revised spelling and grammatical mistaken in this manuscript by MDPI.

Round 2

Reviewer 3 Report

No additional comments